# An Assessment of Biodegradability and Phytotoxicity of Natural Rubber in a Simulated Soil Condition via CO_2_ Evolution Measurement

**DOI:** 10.3390/polym16172429

**Published:** 2024-08-27

**Authors:** Sirichai Pattanawanidchai, Pongdhorn Saeoui, Thanawadee Leejarkpai, Peeraphong Pokphat, Banphot Jiangchareon, Swieng Thuanboon, Nattawut Boonyuen, Chanwit Suriyachadkun, Chomnutcha Boonmee

**Affiliations:** 1National Metal and Materials Technology Center (MTEC), National Science and Technology Development Agency (NSTDA), Khlong Nueng, Khlong Luang, Pathum Thani 12120, Thailand; sirichp@mtec.or.th (S.P.); pongdhor@mtec.or.th (P.S.); thanawl@mtec.or.th (T.L.); peeraphong.pok@mtec.or.th (P.P.); banphot.jia@mtec.or.th (B.J.); 2Royal Thai Navy, Bangkoknoi, Bangkok 10600, Thailand; swieng@hotmail.com; 3National Center for Genetic Engineering and Biotechnology (BIOTEC), National Science and Technology Development Agency (NSTDA), Khlong Nueng, Khlong Luang, Pathum Thani 12120, Thailand; nattawut@biotec.or.th (N.B.); chanwit@biotec.or.th (C.S.)

**Keywords:** biodegradation, rubber, curatives, crosslink, phytotoxicity, soil

## Abstract

In this study, the biodegradation of various natural rubber (NR) samples, i.e., neat NR and NR filled with two different curative contents was investigated under a long-term simulated soil condition at a temperature of 25 ± 2 °C in accordance with ISO 17556. Natural clay loam soil, with a pH of 7.2 and a water holding capacity of 57.6%, was employed. Under controlled test condition both unvulcanized and vulcanized NR samples having low curative content, respectively designated as UNRL and VNRL, exhibited similar biodegradation behaviors to the neat NR. They showed fast biodegradation at the early stage, and their biodegradation rate did not significantly change throughout the test period (365 days). However, for the NR samples having high curative content, respectively called UNRH and VNRH for the unvulcanized and vulcanized samples, a biodegradation delay was observed within the first 130 days. Surprisingly, the UNRH showed a relatively high biodegradation rate after the induction period. At the end of the test, most of the rubber samples (the neat NR, UNRL, VNRL, and UNRH) showed a comparable degree of biodegradation, with a value ranging from 54–59%. The VNRH, on the other hand, showed the lowest degree of biodegradation (ca. 28%). The results indicate that the number of curatives does not significantly affect the biodegradability of unvulcanized NR in the long term, despite the fact that a high curative content might retard microorganism activity at the beginning of the biodegradation process. Apparently, crosslink density is one of the key factors governing the biodegradability of NR. The phytotoxicity of the soils after the biodegradation test was also assessed and represented in terms of seedling emergence, survival rate, and plant biomass for *Sorghum bicolor*. The values of seedling emergence (≥80%), survival rate (100%), and plant biomass of all soil samples were not statistically different from those of the blank soil, indicating the low phytotoxicity of the tested soils subjected to the biodegradation of the rubber samples. Taken as a whole, it can be concluded that the CO_2_ measurement technique is one of the most effective methods to assess the biodegradability of rubbers. The knowledge obtained from this study can also be applied to formulate more environmentally friendly rubber products.

## 1. Introduction

Natural rubber (NR) is a type of natural polymer synthesized by over 2000 plant species. However, the majority of commercial NR products (99% of the world market) are derived from *Hevea brasiliensis* [1]. The NR latex obtained from *Hevea brasiliensis* is a milky white fluid that is found in the form of colloidal dispersion. The composition of fresh latex comprises approximately 25 to 40% *cis*-1,4-polyisoprene, 60% water, and 5% non-rubber components [2]. Statistically, Thailand, Indonesia, and Vietnam are currently the top three countries among the world’s largest NR producers. NR is extensively utilized due to its high elasticity and strength. Generally, unvulcanized NR is not usable due to its poor strength, low stiffness, lack of shape stability, and poor resistance to heat and chemicals [3]. Vulcanization is thus a necessary process to improve the mechanical properties and durability of the final products.

Among the vulcanization systems, sulfur vulcanization is the most widely used due to its cost-effectiveness while providing rubber vulcanizates with better mechanical and dynamic properties. Accelerators and activators are generally added to speed up the vulcanization rate and, hence, reduce vulcanization time. The reaction path of accelerated sulfur vulcanization begins with the reaction between accelerators (e.g., mercaptobenzothiazole or MBT) and sulfur to form monomeric polysulfides (Ac-S_x_-Ac, where Ac is an organic radical derived from the accelerator). The monomeric polysulfides then interact with rubber to form rubber polysulfides (rubber-S_x_-Ac). Finally, the rubber polysulfides react with other rubber molecules, either directly or through an intermediate, to give crosslinks, rubber-S_x_-rubber [4].

At present, the continuous increase in municipal solid waste (MSW) has become one of the most serious environmental problems worldwide. MSW was generated globally at around 2 billion tons in 2020. According to the forecast, the amount of MSW will increase to 2.7 and 3.2 billion tons in 2030 and 2040, respectively, if no appropriate waste management is enforced [5]. Rubber waste is categorized in a group with textiles, wood, leather, and personal hygiene equipment, which represent around 8% of MSW composition. With the advent of new technologies, easily collectable rubber products such as end-of-life tires and scraps from rubber manufacturers can be recycled. However, there are some rubber products, such as rubber bands, rubber threads, and household gloves, that may be disposed of as MSW at the end of their service life. Traditional disposal methods for these wastes, like landfilling or incineration, may pose harm and have negative impacts on ecosystems. Landfilling or open dumping sites are sources of toxic chemicals due to the release of various chemicals from rubber products that might harm plants, wildlife, and humans. The improper handling of rubber waste in landfills causes a breeding ground for mosquitoes and flies, a risk of accidental fires, and long-term pollution of groundwater [6,7,8,9,10]. Burning rubber can cause the accumulation of air pollutants and emissions of toxic gases such as sulfur dioxide (SO_2_), nitrogen dioxide (NO_2_), carbon monoxide (CO), and fine particles [7,8,9,10]. Moreover, hazardous organic compounds (e.g., dioxins, polycyclic aromatic hydrocarbons (PAHs), dibenzofurans, etc.) are carcinogens that directly affect human health [9,10].

Over the past decade, the biological degradation of NR has gained considerable attention. NR can be degraded by the action of microorganisms [11,12,13]. However, the biodegradation process also relies on chemical and physical actions. Many studies have identified microorganisms that have rubber-degrading potential. These microorganisms are divided into two groups, which are clear zone formation and rubber surface adhesion [13,14,15]. Actinomycetes are categorized as surface-adhering microorganisms, known for their adhesive growth and their effectiveness in degrading rubber compared to other rubber-degrading microorganisms [16,17,18]. Normally, soil is an abundant source of microorganisms, including bacteria, fungi, and actinomycetes, which can degrade both natural and synthetic rubbers and utilize them as carbon and energy sources [7,14,19,20].

So far, there are many reports revealing the isolation of rubber-degrading microorganisms and studying their activity in a culture medium [12,16,21,22,23]. Many types of rubber-degrading microorganisms have been studied, such as *Pseudomonas aeruginosa* AL98 [24], *Bacillus* sp. AF-666 [25], co-culture of *Bacillus cohnii*, and *Brevundimonas naejangsanensis* [26]. There are many techniques that are widely used to monitor the degree of biodegradation, including weight loss determination, scanning electron microscopy (SEM), Schiff’s staining, and attenuated total reflectance-Fourier transform infrared spectroscopy (ATR-FTIR) [19,27]. Among these techniques, the measurement of weight loss in soil burial tests is one of the most widely used. The biodegradation of many rubber samples has been monitored by this technique, including sago starch-filled natural rubber latex (SSNRL) [28], poly(lactic acid)-NR blends [29], and composites of thermoplastic starch/natural rubber blends [30]. However, the weight loss measurement in the soil burial test needs special attention to avoid experimental errors taking place during the specimen collection because, as biodegradation proceeds, the rubber surface becomes sticky, and, thus, it is difficult to completely remove the adhered soil particles from the specimen’s surfaces before weighing. Additional errors can be found when the specimens are disintegrated into small pieces, and some fragments of the specimens are not fully collected for the weight loss measurement.

Although the soil burial test is an appropriate technique for evaluating the biodegradation of polymeric materials, biodegradability in soil is a complex process that involves physical, chemical, and biological actions. The differences in environmental conditions, such as temperature, moisture, and soil type, which generally contain different microbial communities [31], in each individual test may reflect different biodegradation rates. In general, biodegradable materials are broken down via the mineralization process by the action of microorganisms. Organic hydrocarbons are converted into carbon dioxide (CO_2_) and water (H_2_O) under aerobic conditions or into a mixture of carbon dioxide (CO_2_) and methane (CH_4_) under anaerobic conditions [32]. Therefore, the measurement of CO_2_ evolution is an important technique to provide the biodegradation rate directly. As previously mentioned, despite their wide use in monitoring the biodegradability of polymers, weight loss measurement, SEM, and FTIR cannot reflect the bioconversion of the polymer’s carbon backbone into metabolic end products (e.g., CO_2_). ISO 17556 has therefore been developed for the determination of the aerobic biodegradability of plastic materials in soil via CO_2_ measurement [33]. Many polymeric materials, including biodegradable plastics [34], thermoplastic starch foams [35], edible sunflower oil [32], and wood plastic composites [36], have been studied. Recently, the method has also been applied to investigate the biodegradation of tire specimens by estimating the amount of CO_2_ evolved through the Sturm test in 14 days [37] and CO_2_ production after 50 days [38].

To the best of our knowledge, the study of rubber degradation by measuring CO_2_ evolution under a long-term soil burial test is very limited. It is therefore the aim of this study to investigate the biodegradation of NR with different crosslink densities under a long-term simulated soil condition by measuring cumulative CO_2_ production in accordance with ISO 17556. Additionally, the phytotoxicity after the rubber biodegradation process was assessed by measuring seedling emergence, survival rate, and biomass of *Sorghum bicolor*.

## 2. Materials and Methods

### 2.1. Materials

Commercial natural rubber (NR; STR 5 L) was purchased from Union Rubber Product Corporation Co., Ltd. (Chachoengsao, Thailand). 2-Mercaptobenzothiazole (MBT, C_7_H_5_SN) was received from Kij Paiboon Co., Ltd. (Bangkok, Thailand). Zinc oxide (ZnO), stearic acid (C_18_H_36_O_2_), and sulfur (S_8_) were purchased from Chemmin Co., Ltd. (Samutprakan, Thailand). Microcrystalline cellulose powder (20 µm) and potassium hydroxide (KOH) were supplied by Merck (Darmstadt, Germany). Potassium dihydrogen phosphate (KH_2_PO_4_), ammonium chloride (NH_4_Cl), and magnesium sulfate (MgSO_4_·7H_2_O) were supplied by KemAus (New South Wales, Australia). Sodium nitrate (NaNO_3_) was supplied by LOBA Chemie Pvt. Ltd. (Mumbai, India). Urea (CH_4_N_2_O) was purchased from Honeywell Fluka (Seelze, Germany). Sorghum seeds were purchased from Pacific Seeds (Thai) Ltd. (Saraburi, Thailand).

### 2.2. Preparation and Characterization of Rubber Compounds and Vulcanizates

To provide rubber vulcanizates with significantly different crosslink densities, two compounding formulas were designed, as shown in Table 1. NRL and NRH represent the formulas having low and high curative (MBT and sulfur) contents, respectively. The rubber compounds were prepared by using a laboratory internal mixer (Brabender Plasticorder 350E, Duisburg, Germany). The initial mixing temperature, fill factor, rotor speed, and mixing time were set at 50 °C, 0.75, 40 rpm, and 8 min, respectively. After the mixing, the rubber compounds were sheeted on a two-roll mill (Labtech model LRM150, Samutprakarn, Thailand) to form thin rubber sheets with approximately a 2.0 mm thickness. The unvulcanized compounds having low and high curative contents were designated as UNRL and UNRH, respectively. Cure behaviors of the rubber compounds, i.e., minimum torque (M_L_), maximum torque (M_H_), torque difference (M_H_–M_L_), scorch time (t_s2_), and optimum cure time (t_c_90), were determined at 160 °C using a moving die rheometer (MDR; TechPro MD+, Cuyahoga Falls, OH, USA), as per ISO 6502-3 [39]. To prepare the vulcanized rubber sheets, the rubber compounds (the UNRL and UNRH) were compression molded using a hydraulic hot press (Wabash Genesis Series, Wabash, IN, USA) under a pressure of 20 MPa at 160 °C for the optimum cure time (t_c_90) as pre-determined from the MDR, and the obtained vulcanized rubbers were designated as VNRL and VNRH, respectively.

After the vulcanization, the basic mechanical properties of the vulcanized rubbers (the VNRL and VNRH) were determined. The hardness was measured by a Shore A durometer (Wallace H-17A, Dorking, UK) following ISO 48-4 [40]. Five measurements were taken on a standard test specimen, and the mean value was reported. The tensile properties were evaluated by using a universal testing machine (Instron 5566, Norwood, MA, USA) at a crosshead speed of 500 mm/min and a load cell of 1 kN in accordance with ISO 37 [41]. Five dumbbell specimens (die type 1) were prepared and tested. The mean value was reported. The crosslink density was determined by the equilibrium swelling method [42]. For each sample, three test specimens, approximately 1 g, were weighed and immersed in 150 mL of toluene for 7 days at room temperature to ensure swelling equilibrium. The swollen specimens were removed from the toluene, blotted using towel paper, and weighed accurately. Since crosslink density is inversely proportional to swelling degree, for simplicity in calculation, the swelling degree was calculated in accordance with Equation (1) and used to represent the degree of crosslink density in this work. Again, the mean value of the three specimens was reported.
(1)$$\mathrm{Swelling}\;\left(\%\right)=\frac{\left(w_{2}-w_{1}\right)\;}{w_{1}}\times 100$$
where $w_{1}$ and $w_{2}$ are the specimen’s weights (g) before and after swelling in toluene, respectively.

### 2.3. Biodegradation Test

#### 2.3.1. Characterization of the Test Materials

Prior to the biodegradation test, the basic characteristics of the NR samples (the neat NR, UNRL, VNRL, UNRH, and VNRH) and a reference (microcrystalline cellulose powder) were evaluated. In this study, the moisture content and total dry solid content were measured by placing the crucibles containing 5 g of the test samples in an air-circulating oven (FED 720, Binder, Tuttlingen, Germany) at 105 °C for 24 h. The volatile solid content was determined by drying the samples in a furnace at 550 °C for 8 h. The total organic carbon content was analyzed by a total organic carbon analyzer (Shimadzu Corp., Kyoto, Japan).

#### 2.3.2. Soil Preparation and Characterization

The test soil was a clay loam soil collected from a sugarcane plantation site in Nakhon Pathom Province, Thailand, and prepared according to ISO 17556 [33]. The soil obtained from the surface layer was sieved through a 10-mesh screen to remove large contaminants (e.g., plant residue, stones, and other inert materials), which may interfere with the test results. The total dry solid content, the volatile solid content, and the total organic carbon were measured using the same method as previously described. The total nitrogen content was measured by the Kjeldahl method (Gerhardt Vapodest 45 s, Gerhardt GmbH & Co. KG, Königswinter, Germany). The water holding capacity (WHC) of the soil was determined by drying the soil in an oven at 105 °C for 24 h until a constant mass was achieved. The obtained dry soil was mixed with an excessive amount of distilled water and then filtered through filter paper. The hydrated soil was allowed to drain for three hours at room temperature and then measured for the percentage of water (by weight) in the wet soil, known as WHC. The final moisture content of the soil was adjusted to 40% of the WHC with a salt solution (0.2 g KH_2_PO_4_; 0.1 g MgSO_4_; 0.4 g NaNO_3_; 0.2 g urea; 0.4 g NH_4_Cl per kg of test soil). The pH and electrical conductivity of the test soil were determined in a soil suspension prepared at a soil-to-deionized water ratio of 1:5 (*w*/*v*). The test soil was properly kept in a sealed container at 4 °C before being used.

#### 2.3.3. Experimental Set-Up and Biodegradation Assay

The biodegradation of the rubber samples in the soil was performed according to ISO 17556, based on the measurement of cumulative CO_2_ production. The rubber sheets were cut into small pieces with dimensions of 3 mm × 3 mm. About 1 g (dry weight) of the samples was mixed with 500 g of dry soil, and the mixture was introduced into each sealed reactor. Blank reactors were set up using only soil without test materials. All reactors were incubated in the dark at 25 ± 2 °C for 365 days. The CO_2_ released due to mineralization was captured in a 50 mL trapping solution of 0.5 N KOH. The amount of released CO_2_ was measured every two days during the first two weeks and every week thereafter until the end of the test by a total organic carbon analyzer (Shimadzu Corp., Kyoto, Japan). The degree of biodegradation ($D_{t}$) was calculated from the cumulative amount of CO_2_ evolved during each measurement using Equation (2):(2)$$D_{t}=\frac{\sum m_{T}-\sum m_{B}}{ThCO_{2}}\times 100$$
where $\sum m_{T}$ is the amount of CO_2_ (mg) evolved in the test reactor at time *t*; $\sum m_{B}$ is the amount of CO_2_ (mg) evolved in the blank reactor at time *t*; and $\;Th{\mathrm{CO}}_{2}$ is the theoretical amount of CO_2_ (mg) evolved by the test or reference materials.

### 2.4. Plant Toxicity Test

To study the phytotoxicity of rubber biodegradation, blank soil (unexposed soil), reference soil (cellulose-exposed soil), and rubber-exposed soils were collected at the end of the biodegradation test. The basic properties of these soils, such as pH, electrical conductivity, moisture content, total dry solid content, total organic carbon content, and nitrogen content, were characterized using the procedures described previously. After the characterization, the soils were used for the plantation of sorghum seeds (*Sorghum bicolor*), listed as one of the standard test species in OECD 208, OECD guidelines for the testing of substances on seedling emergence and growth [43]. Approximately 120 g (dry weight) of all test soils were filled into pots lined with seed germination paper. Ten seeds of *Sorghum bicolor* were properly placed on top of the soil in each pot and covered with perlite. The test was done in three replicates for each test soil. Water was added to the pots to ensure optimal moisture content. The pots were placed in a controlled chamber at 22 ± 10 °C and 70 ± 25% relative humidity. The pots were initially kept in a dark condition for 5 days until the germination rate in the blank soil pots reached 50%. Thereafter, the pots were exposed to repeated cycles of 16 h of light exposure with a light intensity of 350 ± 50 µE/m^2^/s and 8 h of dark exposure for 14 days. Throughout the test, the pots were repositioned randomly to minimize variations in the growth rate of the plants. At the end of the test, the seedling emergence rate, survival rate, and plant biomass were determined. To measure the plant biomass, the test plants were harvested and dried at 60 °C until a constant weight was achieved. The experimental data were expressed as mean values and standard deviations. For statistical analysis, a one-way ANOVA was performed, with a *p* value of 0.05 as a level of significance, to compare the differences in the means of the results.

## 3. Results

### 3.1. Basic Properties of the Rubber Samples

The cure characteristics of the rubber compounds (the UNRL and UNRH) and the mechanical properties and swelling degree of the corresponding rubber vulcanizates (the VNRL and VNRH) are displayed in Table 2. Obviously, the scorch time (t_s2_) and optimum cure time (t_c_90) of the UNRH were considerably shorter than those of the UNRL. This is easily understandable because the UNRH had a considerably higher amount of MBT, which acted as an accelerator to speed up the vulcanization reactions. As the UNRH contained higher amounts of both accelerator (MBT) and vulcanizing agent (sulfur), it showed a greater value of torque difference, indicating a higher crosslink density because it is widely accepted that the torque difference is directly proportional to the crosslink density.

When the UNRL and UNRH were vulcanized, the corresponding rubber vulcanizates, namely the VNRL and VNRH, were obtained and tested. Their mechanical properties and swelling degree are also given in Table 2. Due to its lower curative content, the VNRL had a lower state of cure or crosslink density than the VNRH, as evidenced by the lower torque difference and higher swelling degree values. It is widely accepted that crosslink density is directly proportional to hardness and modulus but is inversely proportional to elongation at break of rubbers. This is the reason why the VNRL showed significantly lower hardness and modulus (stress at 100% strain) with greater extendibility (elongation at break) than the VNRH. The VNRL also exhibited very low tensile strength (~1.7 MPa), compared with the VNRH, because the rubber molecules in the VNRL were lightly crosslinked as evidenced by its extremely high swelling degree (1631%) and, thus, could not effectively transfer stress during the tensile test. The effect of crosslink density on the mechanical properties of the rubber has previously been reported in the literature [42,44,45,46,47].

### 3.2. Biodegradation Results

#### 3.2.1. Basic Properties of the Test Materials

Table 3 shows the fundamental characteristics of the reference and rubber samples. Apparently, the moisture content of all of the rubber samples was very low, ranging from 0.2–0.4%, due to the hydrophobic nature of natural rubber. However, the moisture content of cellulose was much higher, approximately 4.6%, which could be easily explained by the abundance of hydroxyl groups, making it more hydrophilic. Both the cellulose and neat NR had very high volatile solid contents (>99.6%) because they are pure polymers without the presence of any inorganic substances. The volatile solid contents of the compounded rubber samples (the UNRL, VNRL, UNRH, and VNRH) were slightly lower, falling in the range of 96.6 to 96.8%. The results are not beyond expectation, because all these compounded rubbers contained approximately 2.8% (the NRH) and 2.9% (the NRL) of ZnO, which remained as a residue after the volatile solid content test. As cellulose contains carbon (C), oxygen (O), and hydrogen (H) atoms in its molecular structure, the total organic carbon content of the cellulose was approximately 42%, which is significantly lower than that of the rubber samples (80.1 to 84.2%), which contained only carbon and hydrogen atoms.

The basic properties of the soil used for the biodegradation test are tabulated in Table 4. The results reveal that the soil had suitable properties for the test because its pH value (7.2) fell within the range of 6.0 and 8.0, and its WHC (57.6%) was also in the range of 40% and 60%. After the characterization of the soil properties, the moisture content of the soil was adjusted to 40% of the WHC by the addition of an appropriate amount of salt solution to the soil.

#### 3.2.2. Biodegradation of the Rubber Samples

The average cumulative CO_2_ values of the blank cellulose, and all rubbers are shown in Figure 1. The biodegradation degrees of the reference material and the test rubbers were calculated and plotted against time, as shown in Figure 2. According to ISO 17556, the test is considered valid if the biodegradation degree of the cellulose (reference material) is more than 60% and the amount of CO_2_ evolved from the three blank reactors at the plateau phase or at the end of the test is within 20% of the mean value. From the experimental results, the cellulose powder showed ca. 67.7% biodegradation at the end of the test, and the difference in CO_2_ evolved between the three blank reactors was within the requirement. Therefore, the test results were proven to be valid according to the standard requirements.

According to the results in Figure 2, the neat NR biodegraded very quickly during the first 130 days, and its biodegradation rate tended to slow down somewhat thereafter. At the end of the test (365 days), the biodegradation of the neat NR was approximately 57%. Compared with the neat NR, both the UNRL and VNRL had slightly lower biodegradation rates during the first 130 days, but their biodegradation rates were slightly higher afterwards. At the end of the test, the biodegradation degrees of the UNRL and VNRL were 58.3% and 58.9%, respectively. Theoretically, the crosslinking process, which introduces a three-dimensional network into rubber molecules, should retard the biodegradation process. This phenomenon has previously been reported by Tai et al. [48], who studied the biodegradation of starch-polyurethane (PU) flexible films under soil burial conditions. They found that the chemically grafted starch-PU film showed a lower degradation than that of the physically blended starch-PU film, suggesting that the strong crosslinking between starch and PU makes the material more compact and less susceptible to microbial degradation. Surprisingly, at the end of the test, the VNRL exhibited a slightly higher biodegradation than the neat NR and UNRL, despite its lightly crosslinked structure. This could be explained by the inevitable experimental error. During the biodegradation test, both oxygen and water were regularly added to the soil to ensure its aerobic condition and maintain its moisture content. After the addition of water, the soil was stirred to ensure homogeneity. The stirring process might lead to aggregation of the soft and sticky test pieces of the biodegraded neat NR and UNRL, resulting in an interruption of the microbial adhesion on rubber surfaces. On the other hand, the VNRL was less sticky due to its lightly crosslinked structure and, thus, showed less aggregation of the test pieces than the neat NR and UNRL.

The biodegradation curves of the UNRH and VNRH exhibited a special characteristic; i.e., the samples were hardly biodegraded during the first 130 days, called the induction period. However, after this induction period, the UNRH showed a very high biodegradation rate and reached 54.1% biodegradation at the end of the test. Although the VNRH started to biodegrade faster after the induction period, its biodegradation rate was still very low compared with that of the UNRH, which could be attributed to the dense crosslink structure of the VNRH that impeded its biodegradation. At the end of the test, the degree of biodegradation of the VNRH was approximately 28%, which was significantly lower than that of the other rubber samples, whose biodegradation degrees were in a range of 54 to 59%. The results clearly show that crosslink density, especially at high levels, plays a crucial role in the biodegradation process. This is understandable because the biodegradation of sulfur-crosslinked rubber requires an additional process, namely desulfurization, to break the sulfur crosslinks. Bio-desulfurization can be achieved by the action of either one single microorganism or a mixed consortium [49,50,51,52,53,54,55].

Without crosslinks, the biodegradation of rubber molecules commences with the oxidation reactions catalyzed by various enzymes released from rubber-degrading microorganisms. Such reactions cause the breakage of long-chain rubber molecules. The resulting small molecules are then consumed by the microorganisms and converted into various end products such as CO_2_, H_2_O, or CH_4_, depending on environmental conditions [15,18,19]. For crosslinked rubbers, the sulfur linkages between rubber molecules must be cleaved during the biodegradation process. The devulcanization pathway of vulcanized rubber has been proposed in previous studies [56,57]. Initially, the desulfuration enzymes released from the sulfur-degrading microorganisms will cleave the sulfur linkages (rubber-S_x_-rubber) and turn them into thiols (rubber-SH). The thiol groups will be further oxidized and converted into sulfite ions (SO_3_^2−^) and finally sulfate ions (SO_4_^2−^) by the action of three desulfurases, namely DszA, DszB, and DszC. The full mechanism is fully described in the literature [57]. 

It is noted that only the UNRH and VNRH, which contained very high curative content, showed an induction period. The results imply that the curatives used in this work, particularly MBT, might negatively affect the activity of the rubber-degrading microorganisms. MBT has been known as a hazardous substance to microorganisms [58,59,60]. Nonetheless, certain microbes, such as Alcaligenes sp. MH146 strain CSMB1 [61] and Pseudomonas putida HKT554 [62], can use MBT as a source of carbon and nitrogen. It is therefore possible that the high MBT concentration in the UNRH and VNRH could inhibit the growth of rubber-degrading microorganisms on the rubber surface, leading to an induction period while no significant biodegradation of the rubber was observed. Once the amount of MBT is considerably reduced by the action of the MBT-consuming microorganisms, the growth of rubber-degrading microorganisms becomes more pronounced, resulting in an increased biodegradation rate of natural rubber.

The quantitative biodegradation test of natural rubber through CO_2_ production has also been investigated by Basco and Mollea [63]. They monitored rubber degradation under soil burial at room temperature over a period of 236 days and found similar results; i.e., the amount of CO_2_ evolved in the test reactors containing soil and NR samples was higher than that in the blank reactors. However, the biodegradation rate of natural rubber in their work was slightly lower compared with the results in this work at the same test period, probably due to the differences in (1) soil type, (2) microorganism type and content in the test soil, and (3) test conditions. In addition, the difference in sample preparation might be another reason because the samples in this work were cut into smaller pieces, thereby providing a greater specific surface area to interact with microorganisms.

### 3.3. Plant Toxicity

After the biodegradation test, the soils were collected from the test vessels, and their properties were determined, as shown in Table 5. Taken as a whole, the changes in soil properties after the biodegradation were very small. Compared with the original soil (before the biodegradation test), the soils obtained after the biodegradation test became slightly more acidic because the pH values of the soils were reduced from 7.2 to 5.7–6.1. The electrical conductivity (EC) and moisture content of the soils were slightly higher after the biodegradation test, which may be a result of the addition of salt solution to the soil to improve the soil quality at the beginning of the biodegradation test. All of the test soils presented low EC values in the range of 0.66 to 0.73 dS/m, indicating a low concentration of soluble salts, good soil fertility, and no negative impact on plant growth. The total organic carbon content was not greatly affected by the biodegradation. However, the nitrogen content was slightly higher after the biodegradation, which could be attributed to the contribution of nitrogen atoms from MBT.

Figure 3 displays the results of seedling emergence and the survival rate of *Sorghum bicolor*. Based on the guidelines, the test is considered valid if the seedling emergence and the survival of emerged seedlings in the blank soil pots (the control group) are at least 70% and 90%, respectively. In this study, the seedling emergence and the survival of emerged seedlings in the blank soil pots were 93.33% and 100%, respectively. The results confirmed that the test was considered valid.

According to Figure 3, the seedling emergence of *Sorghum bicolor* was higher than 80% in all of the soil samples, indicating that the biodegradation of all of the rubber samples did not affect the seedling emergence of *Sorghum bicolor*. Following the seedling emergence, the survival rate of *Sorghum bicolor* in all of the soil samples was 100%, and there was no significant difference between the blank and all of the test soils. Again, it can be said that the biodegradation of all of the rubber samples had no negative effect on the survival rate of *Sorghum bicolor*.

The appearance of *Sorghum bicolor* at the end of the test (19 days) is shown in Figure 4. There are three pots for each soil sample because the test was carried out in triplicate. Within the same test soil, no distinguishable variation in the *Sorghum bicolor* trees’ growth was observed, indicating good repeatability of the test. In addition, it can be seen from the visual examination that the *Sorghum bicolor* trees planted in the blank soil and the test soils were not different. In all of the pots, there was no sign of phytotoxic effects such as chlorosis, necrosis, wilting, or deformation of the leaves and stems. Similar observations have been revealed by Cheng et al. [64], who studied the effect of biodegradation of the vulcanized natural rubber latex films in a soil compost on spinach’s growth and nutrients. They demonstrated that the biodegradation of the latex films did not significantly affect the spinach’s growth or its nutrients.

Figure 5 displays the biomass (mg per individual tree) of *Sorghum bicolor* at the end of the plant toxicity test. In the control pot (the blank soil), the biomass of *Sorghum bicolor* was 51.6 mg. The biomass of *Sorghum bicolor* in the test soils fell in the range of 45.2 to 56.2 mg. With statistical analysis, the results reveal that there was no significant difference in the biomass of *Sorghum bicolor* between each test soil and the corresponding blank soil. Again, the results confirm that the biodegradation of the rubber samples did not have a negative impact on the biomass of *Sorghum bicolor*.

## 4. Conclusions

This research reveals the biodegradation behaviors of natural rubber (NR) with different curative and crosslink density levels under controlled soil conditions. The degree of biodegradation was determined by measuring the cumulative CO_2_ quantity resulting from the mineralization of rubber-degrading microorganisms. The results reveal that the neat NR biodegraded very quickly from the beginning of the degradation process, and its biodegradation rate was slightly lower towards the end of the test. The NR samples containing a small amount of MBT (the UNRL and VNRL) showed a slower biodegradation rate than the neat NR at the beginning of the test, probably due to the toxicity of MBT on rubber-degrading microorganisms. However, the biodegradation rates of both the UNRL and VNRL were higher after 130 days of the test. Due to the high MBT concentration in the UNRH and VNRH, these samples were not biodegraded during the first 130 days of the test, yielding the induction period for the biodegradation process. Beyond the induction period, when the MBT concentration was significantly reduced by the MBT-consuming microorganisms, both the UNRH and VNRH started to biodegrade. As expected, the VNRH biodegraded much slower than the UNRH due to its dense crosslink structure, which could impede the biodegradation process. At the end of the test (365 days), the VNRH biodegraded only 28%, whereas the other rubber samples (the net NR, UNRL, VNRL, and UNRH) showed almost double the biodegradation level (ca. 54–59%). Clearly, the results reveal that the biodegradability of NR depends greatly on crosslink density. The results from the phytotoxicity test of the soils obtained from the rubber biodegradation reveal that the biodegradation of rubber did not affect the seedling emergence or survival rate of *Sorghum bicolor*. The growth of *Sorghum bicolor* trees was also unaffected by the rubber biodegradation, as evidenced by the insignificant changes in the biomass. Overall, CO_2_ evolution measurement can be an effective method to monitor the progress of rubber biodegradation. This method can be applied to find out the optimum conditions for rubber biodegradation. For future studies, it is of interest to optimize the biodegradation rate of rubbers by investigating the effects of other parameters, such as soil type, microorganism type, and test or treatment conditions.

## Figures and Tables

**Figure 1 polymers-16-02429-f001:**
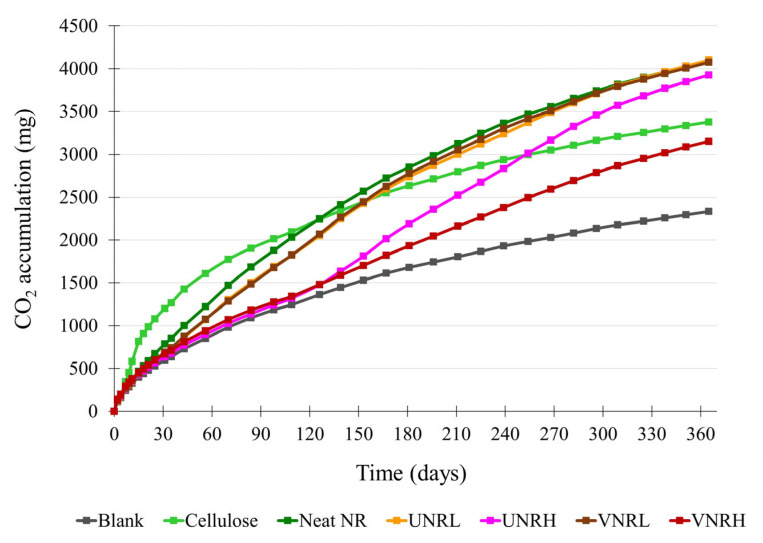
Cumulative CO_2_ productions from blank, cellulose, and test vessels.

**Figure 2 polymers-16-02429-f002:**
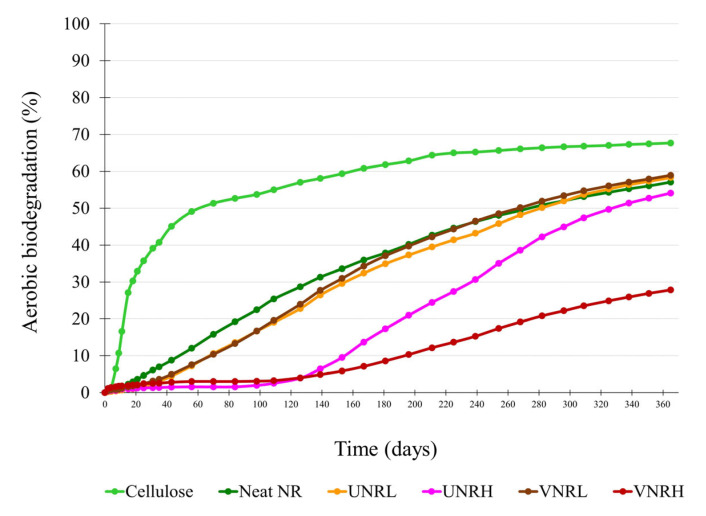
Biodegradation degrees of cellulose and rubber samples.

**Figure 3 polymers-16-02429-f003:**
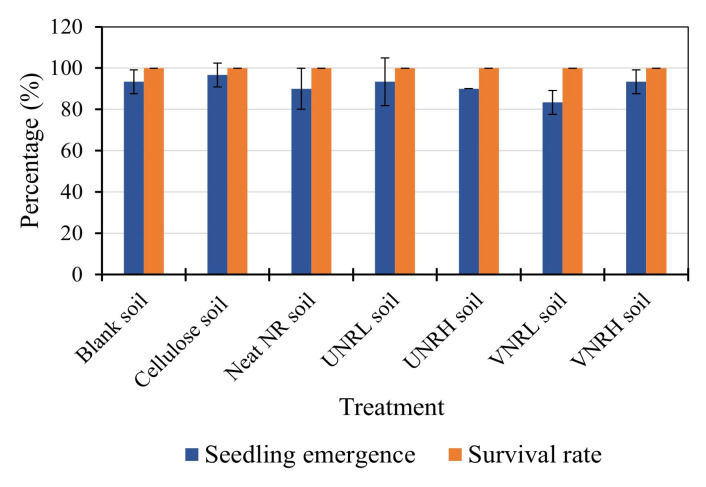
The seedling emergence and survival rate of *Sorghum bicolor* in the test soils.

**Figure 4 polymers-16-02429-f004:**
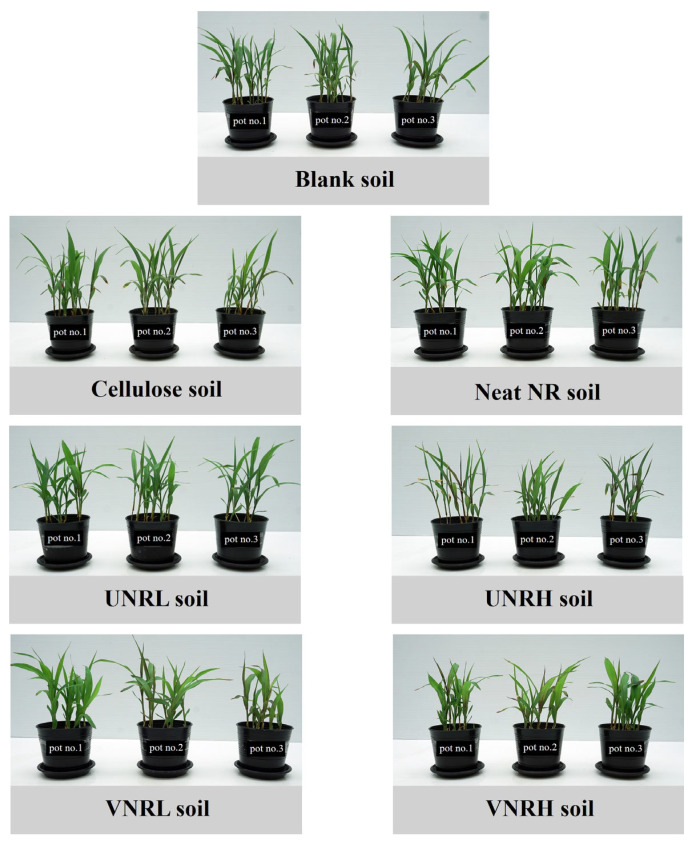
The visual appearance of *Sorghum bicolor* at the end of the test (19 days).

**Figure 5 polymers-16-02429-f005:**
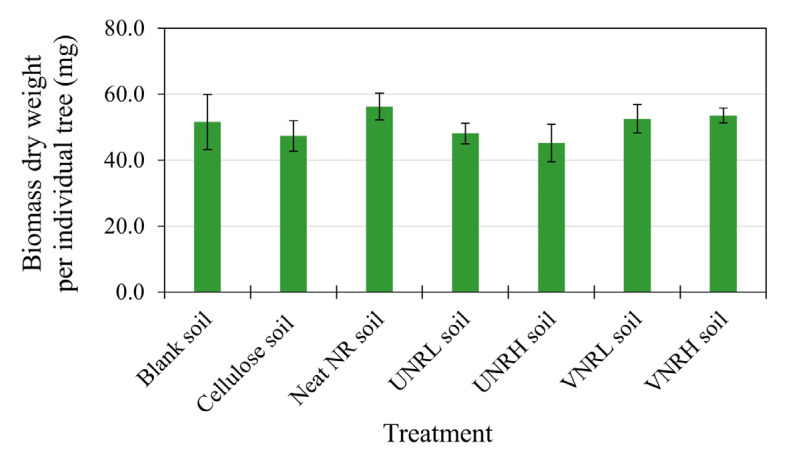
The biomass of *Sorghum bicolor* at the end of the test (19 days).

**Table 1 polymers-16-02429-t001:** The formulas of the rubber compounds used in this study.

Ingredients	Content (Parts per Hundred Rubber; phr)	
	NRL	NRH
Natural rubber (STR 5 L)	100	100
Zinc oxide (ZnO)	3	3
Stearic acid	1	1
Mercaptobenzothiazole (MBT)	0.375	1.875
Sulfur	0.5	2.5

**Table 2 polymers-16-02429-t002:** Basic properties of rubber compounds and rubber vulcanizates.

Properties	Test Materials			
	UNRL	UNRH	VNRL	VNRH
M_L_ (dN·m)	1.31	1.46	-	-
M_H_ (dN·m)	2.23	4.86	-	-
M_H_–M_L_ (dN·m)	0.92	3.40	-	-
t_s2_ (min)	6.82	1.53	-	-
t_c_90 (min)	14.53	3.02	-	-
Hardness (shore A)	-	-	21.0 ± 0.2	35.0 ± 0.4
Stress at 100% strain (MPa)	-	-	0.29 ± 0.01	0.66 ± 0.01
Tensile strength (MPa)	-	-	1.7 ± 0.0	17.6 ± 0.4
Elongation at break (%)	-	-	854 ± 21	726 ± 33
Swelling degree (%)	-	-	1631 ± 2	430 ± 2

**Table 3 polymers-16-02429-t003:** The characteristics of the reference and test materials.

Properties	Test Materials					
	Cellulose	Neat NR	UNRL	UNRH	VNRL	VNRH
Moisture content (% wt.)	4.6 ± 0.0	0.4 ± 0.0	0.4 ± 0.0	0.2 ± 0.0	0.3 ± 0.0	0.2 ± 0.0
Total dry solid content (TS, % wt.)	95.4 ± 0.0	99.6 ± 0.0	99.6 ± 0.0	99.8 ± 0.0	99.7 ± 0.0	99.8 ± 0.0
Volatile solid content (VS, % wt. on TS)	100.0 ± 0.0	99.6 ± 0.0	96.7 ± 0.0	96.7 ± 0.0	96.6 ± 0.0	96.8 ± 0.0
Total organic carbon content (TOC, % wt. on TS)	42.0	84.2	82.5	80.3	80.6	80.1

**Table 4 polymers-16-02429-t004:** The soil characteristics at the beginning of the biodegradation test.

Characteristics	Test Soil
pH	7.2 ± 0.0
Electrical conductivity (dS/m)	0.59 ± 0.05
Initial moisture content (% wt.)	7.2 ± 0.1
Total dry solid content (TS, % wt.)	92.8 ± 0.1
Volatile solid content (VS, % wt. on TS)	3.7 ± 0.0
Water holding capacity (WHC, % wt.)	57.6 ± 7.4
Total organic carbon content (TOC, % wt. on TS)	1.3
Total nitrogen content (% wt. on TS)	0.11

**Table 5 polymers-16-02429-t005:** The basic properties of all of the test soils at the beginning of the plant toxicity test.

Characteristics	Blank Soil	Test Soils					
		Cellulose	Neat NR	UNRL	UNRH	VNRL	VNRH
pH	5.7 ± 0.1	6.0 ± 0.1	5.9 ± 0.1	6.0 ± 0.1	5.8 ± 0.0	6.0 ± 0.1	6.1 ± 0.1
Electrical conductivity (dS/m)	0.67 ± 0.04	0.71 ± 0.03	0.68 ± 0.03	0.71 ± 0.01	0.66 ± 0.03	0.66 ± 0.01	0.73 ± 0.01
Moisture content (% wt.)	11.2 ± 0.2	11.2 ± 0.4	13.2 ± 0.0	11.5 ± 0.5	11.7 ± 0.2	12.3 ± 0.2	10.0 ± 0.2
Total dry solid content (TS, % wt.)	88.8 ± 0.2	88.8 ± 0.4	86.8 ± 0.0	88.5 ± 0.5	88.3 ± 0.2	87.7 ± 0.2	90.0 ± 0.2
Total organic carbon content (% wt. on TS)	1.2	1.3	1.2	1.3	1.3	1.1	1.1
Total nitrogen content (% wt. on TS)	0.35	0.27	0.42	0.31	0.21	0.22	0.22

## Data Availability

The original contributions presented in the study are included in the article, further inquiries can be directed to the corresponding author.

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
