# Peer review of "An Assessment of Biodegradability and Phytotoxicity of Natural Rubber in a Simulated Soil Condition via CO2 Evolution Measurement"

_polymers, 2024, doi:10.3390/polym16172429_

Round 1

Reviewer 1 Report

Comments and Suggestions for Authors

The topic is interesting and relevant. Natural rubber is an important representative of degradable amorphous polymers. The article needs significant refinement of the discussion of the results and the scientific component must be extended too.

Recommendations:

1) In the Abstract, specify how long the experiment lasted in total Introduction: 2) The mechanism and essence of vulcanization should be described in more detail. 3) I recommend giving structural formulas 4) Lines 52-65: The data are for 2013 and 2018, with links to 2018 and 2021. It's 2024 now. It is necessary to update the information, at least in 2019-2022.

Results Table 2 5) How are all these properties set? The methods of the relevant references should be given. I recommend a more detailed discussion of the effect of additives on polymer properties. Section 3.2 6) I recommend giving a graph of biodegradation, as the ratio of mass loss to time, or giving photos of samples. The lack of SEM (microscopy) also makes this work extremely scarce. We can't say anything about the mechanism of the process. There is no data on the structure of materials and its changes. There is no information about how degradation occurs. What is the chemical mechanism? It is worth quoting at least from the literature. Extraction from the soil can also be analyzed or literature data can be provided, why this is not necessary. This section is still not very informative. Section 3.3 7) Fig. 4 requires more discussion, detail and lines. It is unclear if there are differences and what they are. As for the whole article: 8) There is no discussion section. There are only observations, it is important to significantly increase the scientific component of the article.

Author Response

1) In the Abstract, specify how long the experiment lasted in total

Response: We have added the total test duration (line 24) to the revised manuscript, as kindly suggested.

2) In the Introduction, the mechanism and essence of vulcanization should be described in more detail.

Response: We have added the mechanism of vulcanization (lines 56-65) to the revised manuscript, as kindly suggested.

3) In the Introduction, I recommend giving structural formulas.

Response: The chemical formulas of all chemicals used in this work have been added (see the material section) to the revised manuscript, as kindly suggested.  

4) In the Introduction, Lines 52-65: the data are for 2013 and 2018, with links to 2018 and 2021. It's 2024 now. It is necessary to update the information, at least in 2019-2022.

Response: We have updated the information, as shown in lines 66-71.

5) In the results, Table 2: how are all these properties set? The methods of the relevant references should be given. I recommend a more detailed discussion of the effect of additives on polymer properties.

Response: In this study, the rubber compounds and vulcanizates were tested in accordance with relevant ISO standards (mentioned in lines 167, 177, 180), while the biodegradation test of rubber samples was done according to ISO 17556 (line 207). A more detailed discussion of the effect of crosslink density, resulting from the presence of additives, on polymer properties has been added to the revised manuscript (lines 290-297).

6) In the results, Section 3.2 I recommend giving a graph of biodegradation, as the ratio of mass loss to time, or giving photos of samples. The lack of SEM (microscopy) also makes this work extremely scarce. We can't say anything about the mechanism of the process. There is no data on the structure of materials and its changes. There is no information about how degradation occurs. What is the chemical mechanism? It is worth quoting at least from the literature. Extraction from the soil can also be analyzed or literature data can be provided, why this is not necessary. This section is still not very informative.

Response: Thank you for your kind suggestion. According to this study, the degree of rubber biodegradation was continuously monitored by measuring CO2 production, which directly relates to the mineralization of organic carbon in rubber polymers by the action of microorganisms. Unlike the conventional burial tests, which normally determine the weight loss of the specimens, in this study, the rubber specimens were cut into very small pieces, mixed thoroughly with the test soil in the closed reactors, and incubated under a controlled condition as per ISO 17556. Consequently, we could not measure the mass loss or take the specimens for SEM or FTIR determination. However, the mechanism of biodegradation of rubbers has been disclosed in the literature (based on the results from SEM and FTIR) and, thus, we added the biodegradation mechanism to the revised manuscript (lines 386-398), as kindly suggested.

7) In the results Section 3.3, Fig. 4 requires more discussion, detail and lines. It is unclear if there are differences and what they are.

Response: As the test was carried out in three replicates or three pots per one test soil (mentioned in line 252), we therefore put the pictures of all three pots (per one sample) into Fig. 4. However, without good labeling, the unclear figure may cause confusion. We have added the pot number to the figure (Fig. 5 of the revised manuscript) and provided more details as shown in lines 456-458.

8) There is no discussion section. There are only observations, it is important to significantly increase the scientific component of the article.

Response: We have added more discussion to the revised manuscript, as kindly suggested. This includes 1) the brief mechanism of rubber degradation and devulcanization by microorganisms (lines 386-398) and 2) comparison with the results from other researchers that also studied the biodegradation of rubber by CO2 evolution (lines 411-420).

Reviewer 2 Report

Comments and Suggestions for Authors

1. Title

Current title: “Assessment of Biodegradability and Phytotoxicity of Natural Rubber by CO2 Measurement under Simulated Soil Conditions” Suggested revision: “Assessment of Biodegradability and Phytotoxicity of Natural Rubber by CO2 Measurement under Simulated Soil Conditions”  

2. Abstract  

Sentence 1: “This study investigates the biodegradability and phytotoxicity of natural rubber under simulated soil conditions...” Suggestion: Use more precise words to describe the setting of simulated soil conditions, and add details such as temperature, humidity, etc.

Sentence 2: “The CO2 evolution was monitored as an indicator of biodegradation...” Suggestion: Briefly mention why CO2 was chosen as an indicator, and the scientific basis of the method. In the last sentence of the abstract, it is recommended to add the practical significance of the study, such as its contribution to environmental protection or application scenarios.  

3. Introduction  

Background information: Expand the description of the impact of rubber waste on the ecological environment, and cite more statistical data or research literature to support this argument.

Research status: Add the shortcomings of existing rubber degradation technology and explain why existing research cannot fully solve the problem, thereby leading to the research focus of this article.

Research purpose: Emphasize how this study makes up for the above shortcomings, and add a paragraph to specifically explain the innovation of this article, such as the uniqueness of the method or the application prospects of the research.  

4. Experimental design and methods  

Experimental conditions: Describe in detail the basis for selecting simulated soil conditions, such as soil type, pH value, water content, etc., and explain how these conditions correspond to the actual environment.

CO2 measurement method: Add technical details of the CO2 measurement method, such as calibration of the measuring instrument, frequency and time period of data collection.

Data processing: In the data processing section, further explain the statistical methods used and discuss why these methods are most suitable for processing such data.  

5. Results and discussion  

Data display: It is recommended to add more charts to display the data results, such as the trend chart of CO2 release under different soil conditions, or photos of plant growth under different treatment conditions.

Results analysis: In the discussion section, it is recommended to add an in-depth analysis of the results, especially the mechanism behind the data, such as the correlation between CO2 release and rubber decomposition rate.

Comparison with literature: Add comparison with existing literature results to highlight the innovation of this study. For example, discuss the comparison between the degradation rate you found and the degradation rate of different rubber materials in existing studies.  

6. Conclusion  

Research contribution: While summarizing this study, emphasize its practical application value, such as potential impact on rubber waste treatment technology or improvement suggestions.

Future research direction: Add a brief discussion of future research directions, such as how to further optimize the rubber degradation process or other methods to evaluate its environmental impact.

Comments on the Quality of English Language

 Minor editing of English language required.

Author Response

1)  Title, Current title: “Assessment of Biodegradability and Phytotoxicity of Natural Rubber by CO2 Measurement under Simulated Soil Conditions” Suggested revision: “Assessment of Biodegradability and Phytotoxicity of Natural Rubber by CO2 Measurement under Simulated Soil Conditions” 

Response: Thank you for your suggestion. We have changed the title, as kindly suggested.

2)  Abstract

Sentence 1: “This study investigates the biodegradability and phytotoxicity of natural rubber under simulated soil conditions...” Suggestion: Use more precise words to describe the setting of simulated soil conditions, and add details such as temperature, humidity, etc.

Response: We have added more details as shown in lines 18–20 of the revised manuscript.

Sentence 2: “The CO2 evolution was monitored as an indicator of biodegradation...” Suggestion: Briefly mention why CO2 was chosen as an indicator, and the scientific basis of the method. In the last sentence of the abstract, it is recommended to add the practical significance of the study, such as its contribution to environmental protection or application scenarios. 

Response: We totally agree with the reviewer’s comment. However, due to limitations on the number of words in the abstract, we therefore put the requested details in the introduction section (lines 118-123). We have also added the practical significance of the study to the revised manuscript (lines 38-40).

3) Introduction

Background information: Expand the description of the impact of rubber waste on the ecological environment and cite more statistical data or research literature to support this argument.

Response: We have provided more details concerning the impact of rubber waste on the ecological environment, as shown in lines 77-85.

Research status: Add the shortcomings of existing rubber degradation technology and explain why existing research cannot fully solve the problem, thereby leading to the research focus of this article.

Response: Thank you for your suggestions. This point is very important because various techniques widely used to monitor the progress of rubber biodegradation, such as weight loss measurement, SEM, and FTIR, cannot reflect the degree of mineralization of rubber. The CO2 evolution measurement, as per 17556, can provide a direct assessment of the biodegradation of rubber. We have addressed the precautions or shortcomings of the weight loss determination technique, as shown in lines 107-113 of the revised manuscript.

Research purpose: Emphasize how this study makes up for the above shortcomings, and add a paragraph to specifically explain the innovation of this article, such as the uniqueness of the method or the application prospects of the research. 

Response: We have amended the manuscript as shown in lines 133-139. 

4) Experimental design and methods 

Experimental conditions: Describe in detail the basis for selecting simulated soil conditions, such as soil type, pH value, water content, etc., and explain how these conditions correspond to the actual environment.

Response: The soil used in this work was natural soil (clay loam). The soil type has been added to the revised manuscript (lines 19 and 206), as kindly suggested. The basic properties of the soil are given in Table 4 and are in accordance with ISO 17556.

CO2 measurement method: Add technical details of the CO2 measurement method, such as calibration of the measuring instrument, frequency and time period of data collection.

Response: The full details of the method can be found in ISO 17556. The frequency and time period of data collection have been given in lines 231-233.

Data processing: In the data processing section, further explain the statistical methods used and discuss why these methods are most suitable for processing such data. 

Response: For the rubber property test, the number of test specimens was in accordance with the relevant ISO standards. The mean values were reported. Additional details have been added, as shown in lines 177-178,181-183. For the phytotoxicity test, a one-way ANOVA was used in order to compare the differences between blank soil and test soil. The details are given in lines 262-264.

5) Results and discussion 

Data display: It is recommended to add more charts to display the data results, such as the trend chart of CO2 release under different soil conditions, or photos of plant growth under different treatment conditions.

Response: In this study, we carried out the test under one soil condition, and the graph plotted between CO2 release and time is given in Fig. 2. We previously made a mistake by using a plural term for “condition”, leading to misunderstanding, and we have made a correction in the revised manuscript. Similarly, we only carried out the phytotoxicity test under one treatment condition. Thus, we did not have photos of plant growth under different treatment conditions. However, these points are very interesting, and we might carry out the test under different conditions in the future.

Results analysis: In the discussion section, it is recommended to add an in-depth analysis of the results, especially the mechanism behind the data, such as the correlation between CO2 release and rubber decomposition rate.

Response: The correlation between CO2 release and decomposition rate is shown in Equation 2. We have added more information about the devulcanization pathway, as shown in lines 393-398.

Comparison with literature: Add comparison with existing literature results to highlight the innovation of this study. For example, discuss the comparison between the degradation rate you found and the degradation rate of different rubber materials in existing studies. 

Response: We have added more discussion, as shown in lines 411-420 of the revised manuscript.

6) Conclusion 

Research contribution: While summarizing this study, emphasize its practical application value, such as potential impact on rubber waste treatment technology or improvement suggestions.

Response: Additional information has been added to the revised manuscript, as shown in lines 502-504.

Future research direction: Add a brief discussion of future research directions, such as how to further optimize the rubber degradation process or other methods to evaluate its environmental impact

Response: We have addressed the future research direction in the revised manuscript (lines 504-507).

Round 2

Reviewer 2 Report

Comments and Suggestions for Authors

Accept in present form